# Demonstrating change from a drop-in space soundscape exhibit by using graffiti walls both before and after

Martin O. Archer[1, 2], Natt Day[3], and Sarah Barnes[3]

[1]Space and Atmospheric Physics, Department of Physics, Imperial College London, London, UK
[2]School of Physics and Astronomy, Queen Mary University of London, London, UK
[3]Centre for Public Engagement, Queen Mary University of London, London, UK

**Correspondence:** Martin O. Archer
(m.archer10@imperial.ac.uk)

**Abstract.** Impact evaluation in public engagement necessarily requires measuring change. However, this is extremely challenging for drop-in activities due to their very nature. We present a novel method of impact evaluation which integrates graffiti walls into the experience both before and after the main drop-in activity. The activity in question was a soundscape exhibit, where young families experienced the usually inaudible sounds of near-Earth space in an immersive and accessible way. We apply two analysis techniques to the captured before and after data — quantitative linguistics and thematic analysis. These analyses reveal significant changes in participants' responses after the activity compared to before, namely an increased diversity of language used to describe space and altered conceptions of what space is like. The results demonstrate that the soundscape was surprisingly effective at innately communicating key aspects of the underlying science simply through the act of listening. The impacts also highlight the power of sonification in stimulating public engagement, which through reflection can lead to altered associations, perceptions and understanding. Therefore, we show that this novel approach to drop-in activity evaluation, using graffiti walls both before and after the activity and applying rigorous analysis to this data, has the power to capture change and thus short-term impact. We suggest that commonly used evaluation tools suitable for drop-in activities, such as graffiti walls, should be integrated both before and after the main activity in general, rather than only using them afterwards as is typically the case.

## 1 Introduction

Drop-in activities — short, interactive, two-way engagements — tend to form a significant fraction of all non-school public engagement, e.g. $31 \pm 3\%$ of all public activities across the UK's South East Physics Network in 2017/2018 were less than 30 min in duration per individual (Galliano, 2018). Such activities though are difficult to effectively evaluate the impact of, since this necessitates a measure of change on participants (King et al., 2015). While surveys both before and after may be one of the most robust methods of impact evaluation in general (Jensen, 2014), these are neither appropriate for nor commensurate with drop-in activities. This is because participants are arriving all the time, the engagement duration is so short, and surveys risk affecting participants' experience (Grand and Sardo, 2017). A number of evaluation tools more suitable for drop-in activities have been reported including feedback cards, rating cards, snapshot interviews, and graffiti walls (e.g. Grand and Sardo, 2017;

Public Engagement with Research team, 2019). Graffiti walls are large areas (often a wall, whiteboard, or large piece of paper) where participants are free to write or draw responses in reaction to the engagement activity or some prompt question, either directly on the area itself or by sticking responses to it. All of these evaluation methods for drop-ins are particularly useful in process evaluation — assessing the implementation of the activity. Under typical usages (post-activity only) though, they are limited in their ability to routinely demonstrate change from, and thus the impact of, the engagement activity on participants in general.

This paper presents a novel implementation of graffiti walls for impact evaluation, integrating them into both the start and end of a drop-in activity. The activity was a soundscape experience surrounding current space science research that used geo-stationary satellite data converted into audible sound. We show that this evaluation method (through its design, data collection, and analysis) can indeed capture immediate impact — changed language and conceptions of space in this case. Appendices include details of statistical and qualitative coding techniques employed throughout.

## 2  Background

A common misconception is that space is a true vacuum completely devoid of matter and thus there is no activity other than that of the celestial bodies, e.g. planets or asteroids. However, the universe is permeated by tenuous plasmas — gases formed of electrically charged ions and electrons that generate and interact with electromagnetic fields (e.g. Baumjohann and Treumann, 2012). One such example is the solar wind streaming at several hundreds of kilometres a second from the Sun to the edge of the heliosphere, something which only $58 \pm 2\%$ of the UK adult population are aware of (3KQ and Collingwood Environmental Planning, 2015). Space plasmas are also not just limited to our solar system, with other stars having their own stellar winds (e.g. Lamers and Cassinelli, 1999) and the interstellar medium bridging the gap between these plasma bubbles in outer space (Gurnett et al., 2013).

The presence of a medium in space allows for plasma wave analogues to ordinary sound (pressure waves) that occur at ultra-low frequencies — fractions of milliHertz up to 1 Hz. They are routinely measured by many space missions and can have perturbations that are significant fractions of the background values. For a further discussion of the equivalence of these plasma waves to sound see Archer (2020a). One way in which ultra-low frequency waves are generated is through the highly dynamic solar wind buffeting against Earth's magnetic field. This process plays a key role within space weather and thus how phenomena from space can affect our everyday lives (e.g. Keiling et al., 2016). However, the belief by the public that space is completely empty in turn leads many to incorrectly think that there is absolutely no sound in space, reinforced by school science demonstrations such as the bell-jar experiment (see Caleon et al., 2013, for a nuanced discussion of this experiment and sound in near-vacuum conditions) or even popular culture like in the marketing to the movie 'Alien' ("in space no one can hear you scream"). Public engagement with this research area may help correct this fallacy.

Sonification — the use of non-speech audio to convey information or perceptualise data (Kramer, 1994) — can be used to convert satellite measurements of these usually inaudible space sounds into audible signals, simply by dramatically speeding up their playback (Alexander et al., 2011, 2014). This has already been leveraged in public engagement projects for both

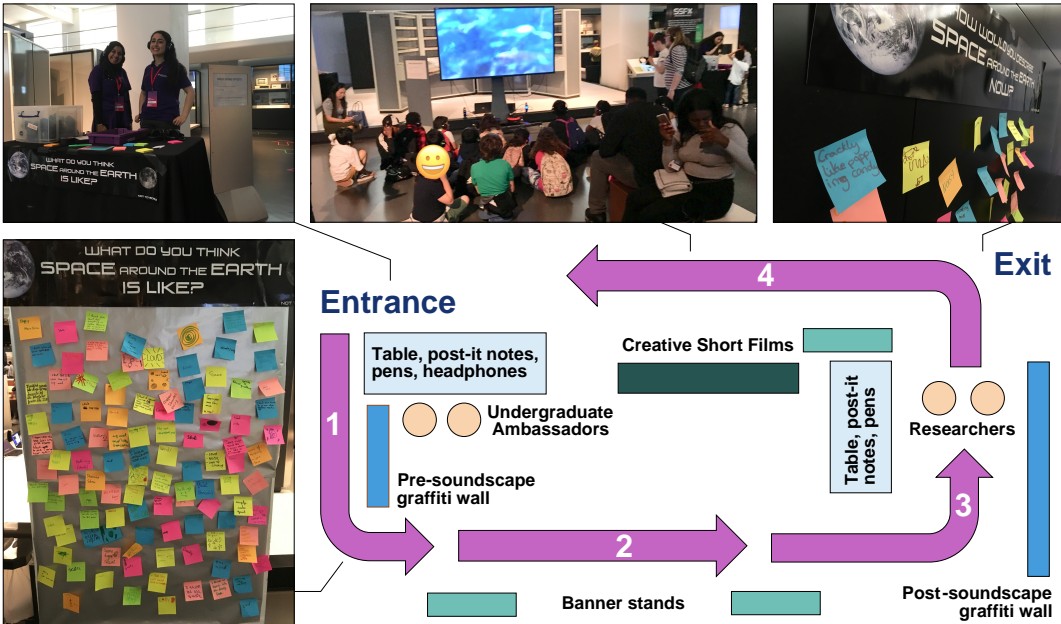

**Figure 1.** Layout and photos of the soundscape exhibit.

scientific and artistic outputs (Archer et al., 2018; Archer, 2020b). Sonification in general has been applied to various scientific datasets (Feder, 2012). Supper (2014) posits that through the public experiencing data in this way it can grip their imagination and produce sublime experiences because of sound's immersive and emotional nature. These arguments, however, are mostly

based on reflections from researchers and artists, rather than through the evaluation of participants' own thoughts and feelings. This paper evaluates the short-term impact on participants of experiencing the sounds of space using graffiti walls both before and after a soundscape.

## 3 Space Soundscape Exhibit

The space soundscape exhibit was held at the free Science Museum in London (United Kingdom) whose informal learning

adopts an inclusive, accessible 'science capital' approach that attracts a diverse range of audiences (Science Museum Group, 2017, 2020). 'Science capital' is defined as the total science-related knowledge, attitudes, experiences, and resources that a person has built up over their life (Archer and DeWitt, 2017). This includes what science they know about, what they think and feel about science, the people they know and their relation to science, and the day-to-day engagement they have with science. The exhibit formed part of the museum's 'Summer of Space Season', held in celebration of the 50th anniversary of the Apollo

moon landings, for which the museum solicited drop-in space-themed activities aimed at young families. It ran between the hours 12:00–16:00 during the May 2019 half-term school holiday over the course of 4 days.

The purpose of the space soundscape was primarily to provide young children and their parents/carers (as key influences upon them) an accessible and immersive experience with space research that would enable participation and spark discussion. Such experiences may, when taken in conjunction with all the other formal and informal interactions with science afforded to a young person, contribute towards developing their science identity and hence build their 'science capital'. Using a generic learning outcomes framework (Hooper-Green, 2004), the main intentions for the activity fell within the realms of 'Enjoyment, Inspiration, Creativity' and 'Attitudes & Values', with explicitly enhancing 'Knowledge & Understanding' being only a secondary aim. Figure 1 shows the layout of the exhibit, which was integrated amongst the museum's usual collections, along with accompanying photos. The activity worked as follows:

1. Museum attendees were invited to participate at the entrance by undergraduate ambassadors. They were first asked to write or draw on a post-it note what they think space around our planet is like. Some younger children required further prompting beyond this broad question however, with ambassadors often asking "what do you think space sounds like?" The participants placed their responses on the pre-soundscape graffiti wall and were handed bluetooth wireless headphones playing the sounds of space.

2. Participants went on a journey while listening to the sounds, following a set of coloured arrows marked out on the floor. A number of banner stands with further information about the sounds were placed along this path, though it was observed that few people read these. This may be either because participants preferred to listen to the sounds or that it was not clear the stands were part of the experience given the exhibit's location amongst other collections.

3. Near the end of the journey, researchers took participants' headphones and asked them to reflect on what they think about space after having listened to the sounds. Participants then recorded their thoughts on post-it notes again and placed these on the post-soundscape graffiti wall. The researchers would use what they had written or drawn to prompt a short dialogue about aspects of the space environment around Earth and space weather research. This method was informed by the 'science capital' research (Archer and DeWitt, 2017), which recommends scientists use and value participants' own experiences within their engagement practice to help enable lower 'science capital' audiences to feel included in science and that science is for "people like me". These discussions provided an opportunity to solidify, or in some cases clarify, the associations that participants made from the soundscape experience in a tailored and audience-focused way (e.g. only going into an appropriate level of detail depending on the individual or group).

4. Finally, researchers would change the channel on the headphones so that participants could watch on a large TV screen a series of creative short films inspired by and incorporating the sounds (Archer, 2020b). The films also featured epilogue text reinforcing the importance and relevance of space weather research. Surprisingly, these artistic films proved much more popular than anticipated.

The graffiti walls were used as an open opportunity for participants to reflect upon their perceptions and associations with space both before and after the soundscape, with this being intentionally left broad to elicit a wide range of possible responses and thus potential impacts. This method was chosen specifically due to its suitability for evaluating drop-in activities, ability to be

integrated within the activity itself, and alignment with our intended overall experience for participants. While graffiti walls are a common evaluation tool, we are unaware of any published public engagement activity that has captured and analysed data both before and after a drop-in activity using them. This makes our evaluation approach for the exhibit novel.

Ethical considerations in the design of the exhibit and its evaluation followed the British Educational Research Association (BERA, 2018) guidelines and were discussed with institutional funders and the Science Museum before the activity occurred. All respondents consented to providing graffiti wall responses as these were not mandatory for participation in the soundscape exhibit. Children only participated in any of the activities when accompanied by their appropriate adult. All data collected was anonymous and no characteristics about participants were solicited. Overall it was deemed (due to the nature of the exhibit, its design, and the types of responses being collected) there was very little risk of harm arising from participation.

The space soundscape was experienced by 1,003 people, recorded using a tally counter. The majority were in family groups (approximately three-quarters were children based on observations) with some independent adults also. It was observed that in families typically only the children contributed to the graffiti walls (with no substantive difference in respondents before and after) and in many cases accompanying adults did not take headphones when offered, perceiving the activity as just for their children. There were 535 and 446 responses (predominantly textual) on the pre- and post-soundscape graffiti walls respectively, rates of $53 \pm 2\%$ and $44 \pm 2\%$. This is some 3–10 times greater than reported for typical graffiti walls (Public Engagement with Research team, 2019) likely due to their integration into the overall activity here.

## 4 Results and Analysis

The data captured on the pre- and post-soundscape graffiti walls are displayed in Figure 2. However, simply presenting these is insufficient to robustly demonstrate any potential changes and thus impacts. Instead, analysis is required and two approaches are taken here, namely quantitative linguistics and thematic analysis.

### 4.1 Quantitative linguistics

Quantitative linguistics investigates language using statistical methods and has uncovered several linguistic laws that mathematically formulate empirical properties of languages. One of these is Zipf's law — the frequency of words are approximately inversely proportional to their rank (where the more often a word is used the higher its rank, i.e. closer to 1) (Zipf, 1935, 1949). An alternative way this law is stated is that the statistical distribution of word ranks follows a power law with an exponent that is typically quoted as $-1$ . Zipf's law holds well for almost all languages as well as many other human-created systems (Piantadosi, 2014). The Zipf exponent, however, can vary and is a measure of the diversity of words. Baixeries et al. (2013) showed that children's Zipf exponents become less-negative / shallower with age, demonstrating increasing variety of language and thus linguistic complexity as they develop. However, we are not aware of Zipf's law being exploited in public engagement evaluation before.

Figure 3 shows rank-frequency plots of the textual responses to the soundscape before and after the experience. This particular analysis thus omits any purely pictorial responses. Ties in ranks have been accounted for using standard competition

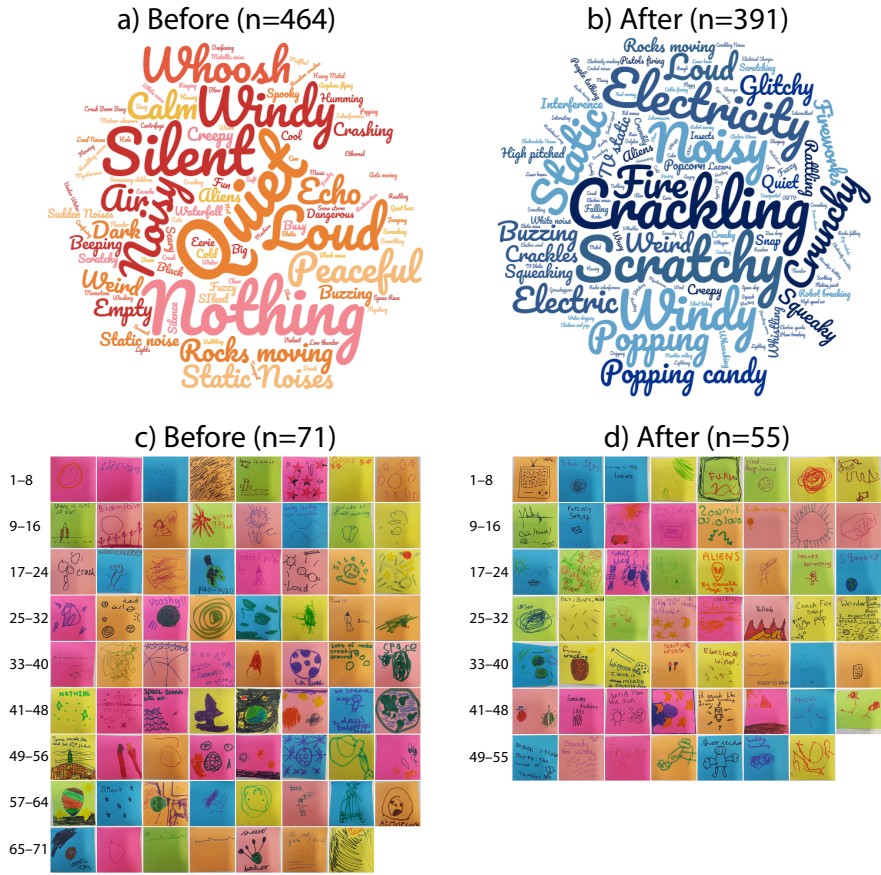

**Figure 2.** Wordclouds (a,b) and drawn images (c,d) from both before (a,c) and after (b,d) experiencing the soundscape.

ranking (also known as "1224" ranking, where a gap is left following the tie). It is clear from these plots that the distributions follow broken power laws (apart from the top word which is of similar frequency before and after). Break points and exponents have been ascertained by a piecewise regression (see Appendix A). Interestingly, the breaks in the two datasets occur at similar ranks namely $\sim$2–3 and $\sim$9–10. We are not concerned with the specific values of the Zipf exponents, which could depend on the demographics of participants, but simply whether they changed from before to after and in what sense. The exponents in the higher rank segments show clear differences — the after dataset exhibits a much shallower exponent. The lowest ranked segments are, in contrast, consistent with one another. The top 10 ranks constitute $62 \pm 2\%$ of words before and $45 \pm 3\%$ after, making the two entire distributions significantly different ($p = 8 \times 10^{-11}$ in a two-sample Kolmogorov-Smirnov test, see Appendix A). The overall effect is an increased diversity of words resulting following the soundscape. We interpret this positive impact as signifying the participants engaged with and reflected on the stimulating experience afterwards, rather than continuing to draw from common associations concerning space which they likely did beforehand. We have therefore

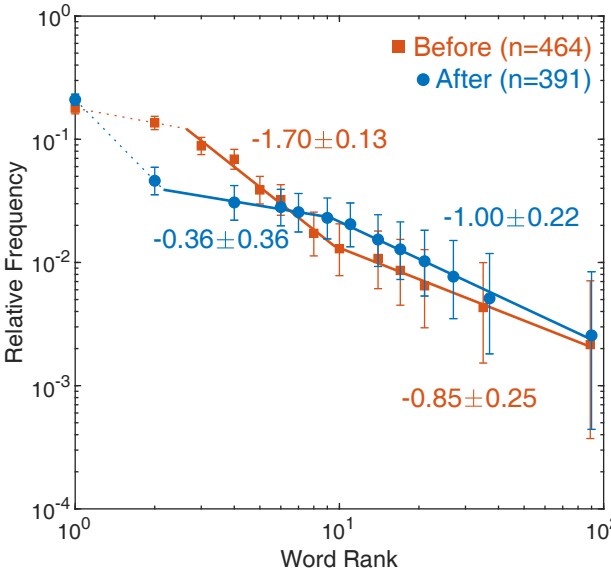

**Figure 3.** Log-log rank-frequency plot of words before (orange) and after (blue) the soundscape. Power law exponents from a piecewise linear regression are indicated. Uncertainties refer to standard errors.

demonstrated language change in participants resulting from a public engagement activity through the novel usage of Zipf's law applied to graffiti wall responses.

## 4.2 Thematic analysis

Thematic analysis (Braun and Clarke, 2006) was used to analyse the meaning behind both textual and drawn responses. This finds patterns, known as qualitative codes, in the data which are then grouped into broader related themes. Instead of using pre-determined codes, the analysis drew on grounded theory (Robson, 2011; Silverman, 2010), allowing the themes to emerge from the data itself as outlined in Appendix B. This more exploratory and data-driven approach enables unexpected outcomes and impacts (be they positive or negative) to come to light, rather than analysing the qualitative data only through a particular lens based on specific intended outcomes. The main themes and underlying (typically antithetical) codes determined by the first author are given in Table 1.

We quantify the number of responses in each theme and qualitative code (cf. Sandelowski, 2001; Sandelowski et al., 2009; Maxwell, 2010) to investigate any changes from before to after the soundscape experience. These are shown in Figure 4 relative to the total responses (panel a) and within each theme (panel b).

The theme of sound is highly relevant to the activity and was commonly expressed both before and after. Responses beforehand mostly considered space to be quiet/silent ($61 \pm 3\%$ within the theme). However, a non-negligible fraction thought it to be loud, which may be due to participants second-guessing the question because of the nature of the activity and/or the phrasing by undergraduate ambassadors. Nonetheless, the overwhelming majority ($97 \pm 1\%$ within the theme) after the experience

| Themes | Codes | Description |
|---|---|---|
| Sound | Quiet | Space is "silent" or "quiet" |
| | Loud | Space is "loud" or "noisy" |
| Emptiness | Empty | Space is an "empty" vacuum with "nothing" in it |
| | Full | Space is filled with material or activity such as "wind" |
| Dynamism | Slow | Space is slow (e.g. "calm" or "peaceful") |
| | Busy | Space is highly dynamic exhibiting busy movement |
| Electricity | Electrical | Expressions of electrical phenomena |
| Space Objects | Space Objects | Commonly known celestial bodies (planets, stars, meteors etc.) or artificial spacecraft |

**Table 1.** Themes and underlying qualitative codes in the thematic analysis.

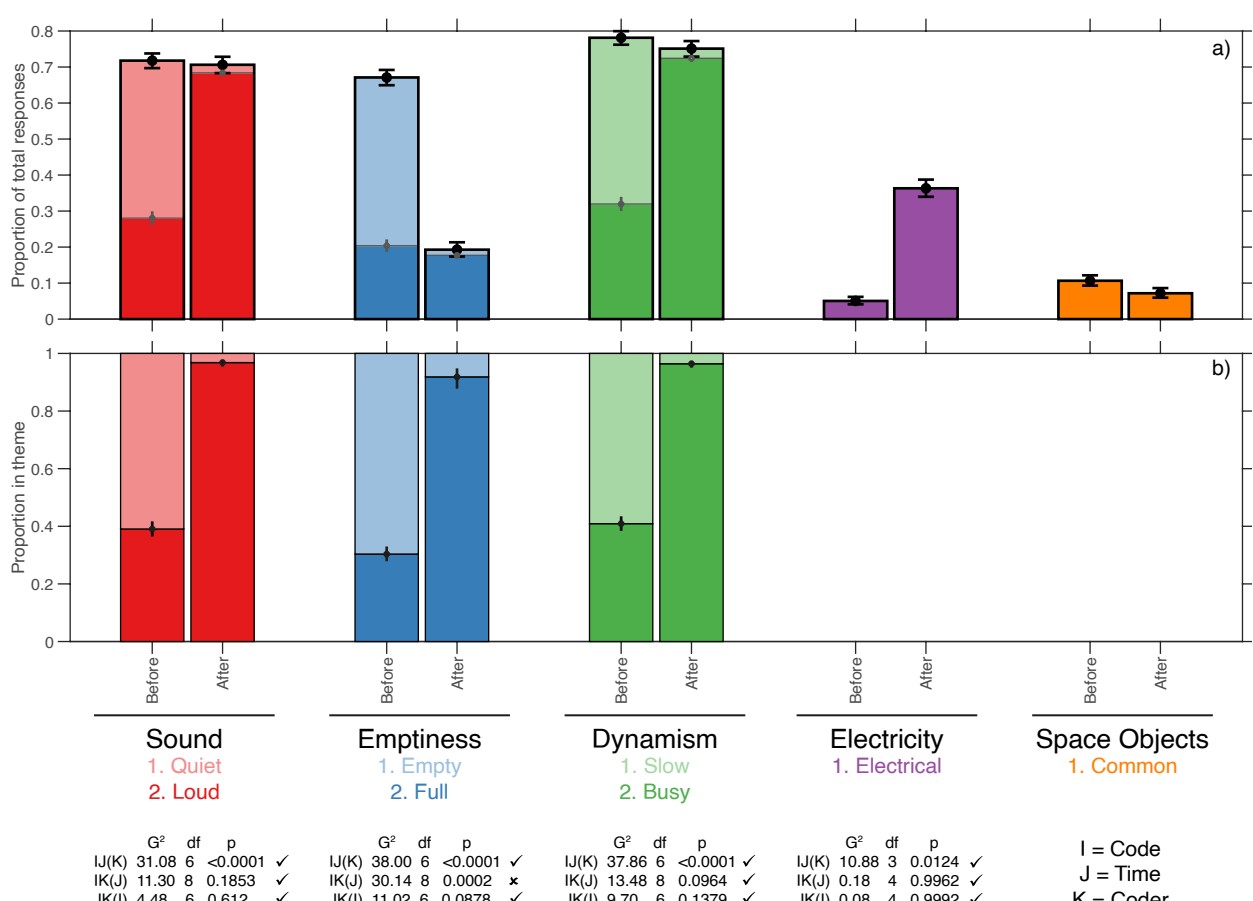

**Figure 4.** Comparison of qualitative themes and codes before ($n = 535$) and after ($n = 446$) the soundscape experience normalised by total responses (a) and totals within each theme (b). Error bars depict the standard error in proportions. Log-linear analysis statistics of the agreement between coders are also shown for each theme.

expressed space to be a noisy environment — a considerable change to beforehand. The perceived loudness of sound, both in terms of human hearing and measurement, necessitates logarithmic scales (Robinson and Dadson, 1956). Such scales, like the decibel, therefore require some reference base-level. For sound this is typically set at the threshold pressure for human hearing of $20\,\mu\text{Pa}$ (Roeser et al., 2007). One must remember though that pressure fluctuations depend on the background pressure level also ($100,000\,\text{Pa}$ at sea level). Therefore, while the absolute amplitude of variations in space are clearly small, relative to the

background they are large (as was noted in section 2) and thus one can consider space to be "noisy" in this sense. Another equally valid perspective is that the process of sonification has revealed the presence of sound that would otherwise not be audible and thus participants have discovered, thanks to the exhibit, that space is "noisier" than they had previously imagined.

    We note that the theme of dynamism exhibits quantitatively similar results to that of sound — a clear majority ($59\pm3\%$ within the theme) thought space to be slow beforehand, whereas the vast majority ($96\pm1\%$) consider it highly dynamic afterwards.

The dynamism of Earth's magnetosphere is relative to the natural timescales of the system. The typical periods of oscillations are of the order of several to tens of minutes, and the properties of the waves (and even their drivers) can significantly change within just a few wave periods (e.g. Keiling et al., 2016). This is unlike most sounds we are used to on Earth, which often remain coherent for many hundreds or even thousands of oscillations. Therefore, just like with sound, space around our planet can be considered dynamic both relative to the properties of the environment and relative to participants' prior expectations.

The theme of emptiness (including both of its underlying codes) was quite common in responses beforehand, however it was expressed much less often following the soundscape. The prevailing opinion before was that space is empty and this dramatically reduced following the soundscape, both relative to the total responses (from $47\pm2\%$ to $2\pm1\%$) and within the theme (from $70\pm3\%$ to $8\pm4\%$). In contrast, the expression of space being full was communicated a similar number of times both before and after. Therefore, participants that had previously thought space was empty typically went on to write words that

fell within a different theme, rather than a response signifying space as being filled with material. Since space is not absolutely devoid of material, being permeated by tenuous plasmas, the exhibit has successfully challenged this common misconception.

    There was a clear increase in the proportion of responses relating to electricity following the event, from $5\pm1\%$ to $36\pm2\%$. Electricity is of fundamental importance to the plasma state, and thus the increased realisation of this by participants is a welcome change resulting from the exhibit.

Common space objects such as planets, stars, or satellites (typically expressed through drawings) may appear at first glance of Figure 2 to be more frequent before the soundscape than after. As a fraction of the total number of responses though, this difference is small and not strictly statistically significant ($p = 0.057$).

    We checked the reliability of all these trends resulting from the qualitative coding by applying log-linear analysis to a subset of the data additionally coded by the co-authors (see appendices for details). Using the notation that $I$ denotes the qualitative

codes, $J$ the time (i.e. before or after), and $K$ the different coders, for the results to be consistent one would expect that the $IJ(K)$ test be statistically significant, constituting the reported trends in codes with time, but the $IK(J)$ and $JK(I)$ interactions should not be, indicating independence from individual coders. These statistics are displayed in Figure 4 for each theme (apart from space objects which was less common) indicating the expected behaviour apart from in the case of emptiness. This theme showed some inconsistency between coders for the "full" code, whereas when only "empty" was considered coders

were in agreement ($G^2 = 32.2, 3.42, 2.06$ respectively). Therefore, the main results of the paper are robust and hence we have demonstrated a change in conceptions of space, well-aligned with the underpinning research, that resulted from this drop-in engagement activity.

## 5  Conclusions

A challenge within public engagement is evaluating the impact of drop-in activities since this necessitates a measure of change using evaluative tools that are appropriate to and commensurate with the engagement (Jensen, 2014; King et al., 2015; Grand and Sardo, 2017). We have presented a novel implementation and analysis stemming from a common evaluation tool, graffiti walls (e.g. Public Engagement with Research team, 2019). These were integrated both before and after a soundscape exhibit on space science research using sonified satellite data. The pre- and post-soundscape graffiti walls provided data on participants' conceptions of space and, through their integration into the activity itself, had much higher response rates than is typical. The captured data was analysed in two different ways.

We investigated the statistical properties of the words expressed by using Zipf's law from quantitative linguistics. This states that the frequency of words in languages typically follow power laws whose exponents give a measure of the diversity of words, where shallower exponents indicate greater variety. The distributions from the graffiti walls showed that the exponent for the top $\sim 10$ words (constituting $62 \pm 2\%$ of the responses before and $45 \pm 3\%$ after) became significantly shallower from before to after, whereas the exponents were consistent for the remaining words. This demonstrates an overall increased linguistic complexity concerning participants' thoughts about space following the activity. This positive result aligns with the exhibit's aims in the realm of 'Enjoyment, Inspiration, Creativity' (cf. Hooper-Green, 2004), since being exposed to the sounds of space led to stimulation, reflection, and ultimately a more diverse and creative set of words about space than had been expressed beforehand. We are unaware of Zipf's law being used in impact evaluation for public engagement before.

We also investigated themes present in the responses, which again yielded significant and robust positive changes from before to after. Beforehand participants typically expressed common misconceptions of space being completely empty, silent, and with little activity. However, after experiencing the space sounds they felt space was a noisy and dynamic environment with electrical phenomena present. It is astounding that simply by listening to the sounds these simple aspects of the underlying space plasma physics were successfully and innately communicated to participants before they even spoke to the researchers. This therefore demonstrates the power of sonification for audiences. While this had been argued by Supper (2014) based on reflections from researchers and artists, here we have shown it from evaluating participants' experiences directly. Therefore, we have shown postive effects in the realms of 'Knowledge & Understanding' and 'Attitudes & Values' (cf. Hooper-Green, 2004) resulting from the soundscape. The measured changes in associations, conceptions, and perceptions will have been further reinforced by researchers drawing from participants' own reflections in their subsequent dialogues (cf. Archer and DeWitt, 2017).

Overall, integrating existing evaluation tools suitable for drop-in engagement activities, such as graffiti walls, both before and after a drop-in activity can enable practitioners to demonstrate changes resulting from the engagement and therefore its

short-term impact. However, typically such tools are only used following activities, which limits the ability to demonstrate some measure of change and thus impact. We suggest that our approach, both in terms of data capture and analysis, should be adopted more regularly, not just for soundscape exhibits, but for a range of different drop-in activities in general.

## Appendix A: Statistical techniques

Statistical uncertainties in proportions are estimated using the Clopper and Pearson (1934) conservative method based on the binomial distribution, where standard (68%) errors are shown throughout.

A piecewise linear regression in log-log space was used to minimise the sum of squared error between the data and a model made up of a specified number of line segments whose break points could be varied iteratively. This was performed for an increasing number of segments, each time calculating the degrees-of-freedom-adjusted $R^2$ which accounts for the number of explanatory variables added to the model:

$$\overline{R}^2 = 1 - \left(1 - R^2\right) \frac{n-1}{n-m-1} \tag{A1}$$

where $R^2$ is the usual coefficient of determination, $n$ is the number samples, and $m = 2s - 1$ is the total number of explanatory variables in the piecewise linear model with $s$ segments. The final model was selected as the first peak in $\overline{R}^2$ with $s$. Any segments with only two datapoints are later ignored. The statistical significance of the slopes was determined by ANCOVA with a multiple comparison procedure (Hochberg and Tamhane, 1987). The standard errors in the slopes quoted are derived from a propagation of uncertainty in the proportions within the linear regression.

A two-sample Kolmogorov-Smironov test is used to non-parametrically test the equality of two probability distributions. It quantifies the distance between two one-dimensional empirical (cumulative) distribution functions $F_{1,n}(x)$ and $F_{2,m}(x)$ as

$$D_{n,m} = \sup_x |F_{1,n}(x) - F_{2,m}(x)| \tag{A2}$$

where $\sup$ is the supremum function (Massey, 1951). The critical value of this statistic is given by $\sqrt{-1/2 \ln(\alpha/2)(m+n)/mn}$ for desired significance $\alpha$.

Finally, log-linear analysis is employed to check the consistency of the changes in coding with time across the different coders. This extension of the $\chi^2$ test of independence to higher dimensions uses a similarly distributed statistic, the deviance, given by

$$G^2 = 2 \sum O_{ijk} \ln \frac{O_{ijk}}{E_{ijk}} \tag{A3}$$

for observed $O_{ijk}$ and expected $E_{ijk}$ frequencies (Agresti, 2007). Here we assess conditionally independent models denoted $IJ(K)$, which tests the two-way $IJ$ interaction with the effects of the $IK$ and $JK$ interactions removed. Computationally this calculates $G^2$ for each level of $K$ summing the results, with $G^2$ having $(n_I - 1)(n_J - 1)n_K$ degrees of freedom.

| Theme | Codes | Before | | After | |
|---|---|---|---|---|---|
| | | Unique | Total | Unique | Total |
| | | $(n = 202)$ | $(n = 535)$ | $(n = 190)$ | $(n = 446)$ |
| Sound | 1. Quiet | 22 | 234 | 7 | 10 |
| | 2. Loud | 59 | 150 | 91 | 305 |
| | Total | 81 | 384 | 98 | 315 |
| Emptiness | 1. Empty | 36 | 250 | 4 | 7 |
| | 2. Full | 31 | 109 | 37 | 79 |
| | Total | 67 | 359 | 41 | 86 |
| Dynamism | 1. Slow | 33 | 247 | 8 | 12 |
| | 2. Busy | 74 | 171 | 111 | 323 |
| | Total | 107 | 418 | 119 | 335 |
| Electricity | 1. Electrical | 13 | 27 | 36 | 162 |
| Space Objects | 1. Common | 51 | 57 | 154 | 32 |

**Table B1.** Number of responses (both unique and total) in each theme before and after the soundscape.

## Appendix B: Qualitative coding

The qualitative coding process of thematic analysis drawn from grounded theory involved the following steps:

1. Familiarisation: Responses (Figure 2) are studied and initial thoughts noted.

2. Induction: Initial codes are generated based on review of the data.

3. Thematic Review: Codes are grouped together into themes and applied to the full data set.

4. Reliability: Codes are applied to a subset of data by second coders to check reliability of results.

5. Finalisation: Theoretical interpretation and narrative are formulated from final coding.

Table B1 shows the number of responses (both unique and total) across words and pictures in each theme and its underlying codes both before and after the soundscape experience. To ensure the reliability of the main qualitative coding of the entire dataset, second coders applied the thematic analysis to a subset of the data. This subset constituted the top 16 words before (58% of total responses) and 15 words after (49%), with the slightly different number of words used in the two datasets being due to ties in the ranking of words making it impossible to have exactly the same number in both. Table B2 shows the totals of how these unique words were grouped across all three coders. These results are used in the log-linear analysis to test reliability, which we note does not require equally sized datasets. The codes' association to the raw data can be found in the supplementary material, both for the main and second coders.

| | | Coder 1 | | Coder 2 | | Coder 3 | |
|---|---|---|---|---|---|---|---|
| | | Before | After | Before | After | Before | After |
| Sound | 1 Quiet | 8 | 0 | 5 | 0 | 8 | 1 |
| | 2 Loud | 6 | 11 | 5 | 11 | 4 | 5 |
| | None | 2 | 4 | 6 | 4 | 4 | 9 |
| Emptiness | 1 Empty | 8 | 0 | 6 | 0 | 9 | 1 |
| | 2 Full | 5 | 3 | 0 | 1 | 6 | 7 |
| | None | 3 | 12 | 10 | 14 | 1 | 7 |
| Dynamism | 1 Slow | 7 | 0 | 5 | 0 | 6 | 0 |
| | 2 Busy | 7 | 11 | 4 | 12 | 10 | 11 |
| | None | 2 | 4 | 7 | 3 | 0 | 4 |
| Electricity | 1 Electrical | 2 | 6 | 2 | 7 | 2 | 6 |
| | None | 14 | 9 | 14 | 8 | 14 | 9 |

**Table B2.** Statistical comparison of the number of unique words in each qualitative code as judged by different coders across a subset of the data (the top 16 words before and 15 words after).

*Data availability.* Data supporting the findings are contained within the article and its supplementary material.

*Author contributions.* MOA conceived the project and its evaluation, performed the analysis, and wrote the paper. ND and SB assisted with the analysis.

*Competing interests.* The authors declare that they have no conflict of interest.

*Acknowledgements.* We thank the researchers (Alice Giroul, Christopher Chen, Emma Davies, Jesse Coburn, Joe Eggington, Luca Franci, Oleg Shebanits) and undergraduate ambassadors (Avishan Shahryari, Cheng Yeen Pak, Christopher Comiskey Erazo, Habibah Khanom, Safiya Merali, Yinyi Liu) who helped deliver the exhibit along with all the staff at the Science Museum (including Becky Carlyle, Imogen Small, Sevinc Kisacik). This project has been supported by QMUL Centre for Public Engagement Large 2016 and Small 2019 Awards, an EGU Public Engagement Grant 2017, and STFC Public Engagement Spark Award ST/R001456/1. M.O. Archer holds a UKRI (STFC / 285   EPSRC) Stephen Hawking Fellowship EP/T01735X/1.

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
