# Peer review of "Demonstrating change from a drop-in space soundscape exhibit by using graffiti walls both before and after"

_Geoscience Communication, 2020_

## Referee Comment (RC1) · John Coxon (Referee) · 19 Oct 2020

In general I enjoyed this manuscript and feel it would be suitable for publication in Geoscience Communication with some changes which I outline below.

95–99: I was unfamiliar with Zipf's law, and I think applying this to evaluate public engagement in this manner is highly novel and very impressive. The less negative the exponent, the wider the vocabulary. In Figure 3, this clearly demonstrates that for the ~10 highest ranks, there was a significantly wider range of vocabulary. However, I do

[Figure]

have some concerns about things that are unclear. Firstly, I think it is necessary to express and discuss the percentage of the words captured in the ~10 highest ranks; from an inspection of the graph it looks to me like ~90% for the "before" set and ~80% for the "after" set, so the vast majority of the language used on the walls is presumably contained in those 10 upper ranks (if I'm misinterpreting this graph, quoting these numbers explicitly is even more important). Secondly, the authors say "While the exponents in the lowest ranked segments are consistent with one another," but the two exponents are not consistent with one another; -1.00±0.04 ≠ -0.85±0.05. This part needs to be rewritten to acknowledge that a) although the two are similar, in fact the variety of language at rank > 10 decreased after the study compared to before; and b) what implications this has.

105: The authors say that "instead of using pre-determined qualitative codes, the analysis drew on grounded theory". I am unfamiliar with the word "code" used in this context. Appendix B particularly, but also the citations, helped me grasp what the meaning is (i.e., for each word, identifying its theme and then within that theme whether it means one of two extremes, i.e. "quiet" or "loud", "empty" or "full", and "slow or busy"). However, I am concerned that readers might not be as tenacious in reading around the terminology as I was as a reviewer, and so I think the paper would be much improved with a fuller explanation here.

108–112: Tying into my criticisms above, a better way to express this would be a table which explicitly outlines which of these are themes and which are codes, I think.

113: I had not heard of Cronbach's alpha prior to this manuscript. Most of the examples I have found of its use during my reading for this review refer to applying it to Likert-scaled quantities to compare the extent to which, say, a questionnaire of questions about a given theme are telling the author about that theme. This indicates two things; firstly, to apply it the themes must have been assigned numerical values, but it is not clear to me what those values were from the explanation herein. If the authors use Cronbach's alpha, they need to explain in more detail how the themes were quantised

in order to apply the measure (are these simply the 1/2 numbers expressed in Figure 4? If so, say that, and if not, then what are they?). Secondly, as far as I can tell from my exploration of the literature, Cronbach's alpha is applied to measure reliability, not dimensionality (as in this study); in fact, some literature I read specifically cautioned against the latter and the Cho paper cited does not seem to provide a rationale for using the measure in this way. I have no real issue with the central analysis tenet in Section 4.2, namely that the coder can differentiate to which theme a word belongs. However, if I did have an issue with it, the quoted statistic would not convince me, and should either be removed or the reasons for its validity should be expanded on significantly.
* * *

---

## Short Comment (SC1) · 20 Oct 2020

We thank Dr Coxon for their time in promptly assessing the manuscript and have considered their comments carefully.

**In general I enjoyed this manuscript and feel it would be suitable for publication in Geoscience Communication with some changes which I outline below.**

**95–99: I was unfamiliar with Zipf's law, and I think applying this to evaluate public engagement in this manner is highly novel and very impressive. The less**

**negative the exponent, the wider the vocabulary. In Figure 3, this clearly demonstrates that for the ∼10 highest ranks, there was a significantly wider range of vocabulary. However, I do have some concerns about things that are unclear. Firstly, I think it is necessary to express and discuss the percentage of the words captured in the ∼10 highest ranks; from an inspection of the graph it looks to me like ∼90% for the "before" set and ∼80% for the "after" set, so the vast majority of the language used on the walls is presumably contained in those 10 upper ranks (if I'm misinterpreting this graph, quoting these numbers explicitly is even more important).**

The proportion of words captured within a segment is the sum of the relative frequencies within that segment, which is similar to the area under the graph. We thank the reviewer for their suggestion to include the percentages in each segment, which we shall include in the revised manuscript and were as follows:

| Rank | Before | After |
|------|--------|-------|
| $\leq 10$ | $62 \pm 2\%$ | $45 \pm 3\%$ |
| $> 10$ | $38 \pm 2\%$ | $55 \pm 3\%$ |

**Secondly, the authors say "While the exponents in the lowest ranked segments are consistent with one another," but the two exponents are not consistent with one another; $-1.00 \pm 0.04 \neq -0.85 \pm 0.05$. This part needs to be rewritten to acknowledge that a) although the two are similar, in fact the variety of language at rank > 10 decreased after the study compared to before; and b) what implications this has.**

This statement was based on the fact that the two segments' error bars overlapped each other's regression lines, however, the reviewer's point has raised the fact that we did not propagate through these errors in the proportions into our standard error in the slope. Doing this we arrive at the following:

| Rank | Before | After |
|---|---|---|
| $\leq 10$ | $-1.70 \pm 0.13$ | $-0.36 \pm 0.36$ |
| $> 10$ | $-0.85 \pm 0.25$ | $-1.00 \pm 0.22$ |

This now better illustrates that the exponents in the lower ranked segments are indeed consistent with one another as well as the standard Zipf exponent of $-1$. Thus the implication is that there was significantly increased diversity of language in approximately half the dataset, whereas the other half were consistent with one another, thereby making the overall effect positive.

**105: The authors say that "instead of using pre-determined qualitative codes, the analysis drew on grounded theory". I am unfamiliar with the word "code" used in this context. Appendix B particularly, but also the citations, helped me grasp what the meaning is (i.e., for each word, identifying its theme and then within that theme whether it means one of two extremes, i.e. "quiet" or "loud", "empty" or "full", and "slow or busy"). However, I am concerned that readers might not be as tenacious in reading around the terminology as I was as a reviewer, and so I think the paper would be much improved with a fuller explanation here.**

We thank the reviewer for this suggestion. We will now include the sentence

> This finds patterns, known as qualitative codes, in the data which are then grouped into broader related themes.

after the introduction of thematic anlaysis. We shall also highlight that in our scheme the codes within a theme are typically antithetical.

**108–112: Tying into my criticisms above, a better way to express this would be a table which explicitly outlines which of these are themes and which are codes, I think.**

We agree with the reviewer and will add the following table:

| Themes | Codes | Description |
|---|---|---|
| Sound | Quiet | Space is "silent" or "quiet" |
| | Loud | Space is "loud" or "noisy" |
| Emptiness | Empty | Space is an "empty" vacuum with "nothing" in it |
| | Full | Space is filled with material or activity such as "wind" |
| Dynamism | Slow | Space is slow (e.g. "calm" or "peaceful") |
| | Busy | Space is highly dynamic exhibiting busy movement |
| Electricity | Electrical | Expressions of electrical phenomena |
| Space Objects | Space Objects | Commonly known celestial bodies or artificial spacecraft |

**113: I had not heard of Cronbach's alpha prior to this manuscript. Most of the examples I have found of its use during my reading for this review refer to applying it to Likertscaled quantities to compare the extent to which, say, a questionnaire of questions about a given theme are telling the author about that theme. This indicates two things; firstly, to apply it the themes must have been assigned numerical values, but it is not clear to me what those values were from the explanation herein. If the authors use Cronbach's alpha, they need to explain in more detail how the themes were quantised in order to apply the measure (are these simply the 1/2 numbers expressed in Figure 4? If so, say that, and if not, then what are they?). Secondly, as far as I can tell from my exploration of the literature, Cronbach's alpha is applied to measure reliability, not dimensionality (as in this study); in fact, some literature I read specifically cautioned against the latter and the Cho paper cited does not seem to provide a rationale for using the measure in this way. I have no real issue with the central analysis tenet in Section 4.2, namely that the coder can differentiate to which theme a word belongs. However, if I did have an issue with it, the quoted statistic would not convince me, and should either be removed or the reasons for its validity should be expanded on significantly.**

The reviewer raises valid concerns on the regular usage of Cronbach's alpha within qualitative research in general and we appreciate that due to the word limit of the GC letters format its usage was not clearly discussed. Given that Cronbach's alpha is not
Interactive comment

critical to the methods or results in the manuscript, we feel it best to simply remove this sentence.

---

## Editor Comment (EC1) · Mathew Stiller-Reeve (Editor) · 5 Nov 2020

During a recent virtual writing retreat, we used a peer-review framework to review your abstract. We then had an open discussion and noted down all the feedback. We also reviewed your abstract with the following advice in mind: "The abstract is a condensed and concentrated version of the full text of the research manuscript. It should be sufficiently representative of the paper if read as a stand-alone document". We looked for the important elements that we believe should be in a research abstract and we comment on them below. We hope the following is helpful for your revisions.

[Figure]

Overall: We were really interested in your project, this "new approach" of graffiti walls and how you have evaluated them. The graffiti walls are a fun method, and we also really liked how you link methods for analysing vocabulary and illustrations. The word "cool" came up a lot when we discussed your project!

The Abstract contained all the necessary parts, which was very positive. You might want to consider tightening up or re-focussing some of these parts to make the Abstract clearer and more concise.

Title: The title contains a lot of information which is great. However, we hope you can make it shorter and more concise. It seems to put more emphasis on the evaluation analyses you used rather than the innovation of the graffiti walls themselves. Several in the group had to read the title several times to pick up on the message. A couple of people misunderstood and thought that the graffiti wall was within the soundscape itself, and that you tested the soundscape before and after the wall was graffitied on.

Need and relevance: The first sentence conveys the need and relevance of the research. However, please consider editing so that it is clearer. Maybe consider two sentences.

Question/hypothesis: Here we misunderstood whether the focus was the graffiti walls being the evaluation itself or the evaluation of the graffiti walls. Right now, it seems that the evaluation methods of quantitative linguistics and thematic analysis could be the main focus of the research question. However, we feel that the graffiti walls themselves are meant to be the main issue. This confusion probably comes from the use of "analysis", "evaluation", "techniques" and "method" in rather quick succession. You might want to look into this to ensure the focus of your main research question/objective is clearly conveyed.

Methods: The methods used for evaluating the graffiti walls are interesting and novel. It's good that you state both methods in the Abstract, but it's not clear what the methods actually do. This part gets quite confusing since it is technically contained in a 5-line

Interactive
comment
sentence, with several clauses. You might consider splitting the methods and results more clearly, and not containing so much detail.

Results: Your main results should refer back to whether the graffiti walls themselves functioned as an evaluation tool. You may also want to refer back to the aim of the whole exhibit. Yes, the graffiti walls may show change, but do they show change relevant to the aim of the actual exhibit itself? Again, the text gets a little complex at this point especially where you write "dynamism, emptiness and electricity, areas highly relevant to the underlying space plasma physics of the sonified data". Our group got a little lost here and asked whether this level of technicality was necessary in the Abstract. You might consider referring to these terms in the main text where you have more room to explain.

Take-home message: You have some really nice results here, that we think you can convey stronger in the final two sentences. Refer to the graffiti walls specifically again and what potential they have. By writing "more broadly" makes it sound like they are already being used in certain circles. Do you mean "more broadly" within soundscapes exhibits or "more broadly" for all drop-in activities in science communication?

Clarity: When it comes to clarity and conciseness, we would like to ask you to consider whether there are redundant words in the Abstract that you could delete. We would also like you to try and split some of the complex sentences to help with the flow of the story.

Spelling/grammar: We all had an issue with the use of "pre- and post-X" with no noun after, in both the title and the text. We're pretty sure you did not mean pre- and post-graffiti walls. If you meant pre- and post-activity graffiti walls, that makes more sense. But you may want to simply use "before and after" as you do later in the text. That's much easier for the reader (us) to relate to.

Again, this seems like a really innovative and exciting project. We hope our comments and suggestions help to make the Abstract even better.

Kind regards, Mathew Stiller-Reeve and several members of The Norwegian Research School for Dynamics and Evolution of Earth and Planets (DEEP)

Please also note the supplement to this comment:
https://gc.copernicus.org/preprints/gc-2020-41/gc-2020-41-EC1-supplement.pdf

---

## Author Comment (AC1) · 17 Nov 2020

We thank Dr Stiller-Reeve and the members of The Norwegian Research School for Dynamics and Evolution of Earth and Planets (DEEP) for their review of our manuscript's title and abstract. It raised several helpful points which we have taken into account to improve not only these aspects, but the entire manuscript.

We have simplified the title to "**Demonstrating change from a drop-in space sound-scape exhibit by using graffiti walls both before and after**".

[Figure]

We have also revised the abstract to the following:

Impact evaluation in public engagement necessarily requires measuring change. However, this is extremely challenging for drop-in activities due to their very nature. We present a novel method of impact evaluation which integrates graffiti walls into the experience both before and after the main drop-in activity. The activity in question was a soundscape exhibit, where young families experienced the usually inaudible sounds of near-Earth space. We apply two analysis techniques to the captured before and after data — quantitative linguistics and thematic analysis. These analyses reveal changes in participants' responses after the activity compared to before, namely an increased diversity of language used to describe space and altered conceptions of what space is like. The results demonstrate that the soundscape was effective at innately communicating aspects of the underlying science. Therefore, we show that this novel approach to drop-in activity evaluation, using graffiti walls both before and after the activity, has the power to capture change and thus short-term impact. We suggest that commonly used evaluation tools suitable for drop-in activities, such as graffiti walls, should be integrated both before and after the main activity in general, rather than only using them afterwards as is typically the case.

Similar changes have been made throughout the manuscript as well.

---

## Referee Comment (RC2) · Anonymous Referee #2 · 18 Nov 2020

\*General comments\*

This is a good paper that presents a useful approach to evaluating drop-in public engagement activities. The detailed statistical analysis is particularly interesting, perhaps more for its explication of a rigorous analysis of graffiti walls and word clouds than for its demonstration of the efficacy of this particular activity. The analysis is very impressive and this paper stands to be a constructive best-practice guide for other public engagement practitioners.

[Figure]

Nevertheless, I think the novelty of using before-and-after graffiti walls is perhaps over-stated. For example, I was part of an interactive drop-in exhibit in March 2018, where we asked attendees to write words / ideas related to the exhibit theme on small cards, both on entry and just prior to leaving, giving us both pre- and post- data, in the form of collections of words and phrases, in much the same way. However, I think the sub-sequent analysis of data performed here is what makes this work noteworthy, and, as far as I am aware, original.

The title adequately reflects the contents of the paper, and the abstract gives a neat summary too. Overall, the paper is well-structured and clear, and of an appropriate length for the material covered. The language is fluent and precise, although there are one or two points (as noted in the specific comments below) where the readability drops off a bit and it becomes confusing. Nevertheless, this paper is largely well-written, useful and enjoyable to read. It makes a worthwhile contribution to the literature of this field.

*Specific comments*

- Line 20: Is it worth explaining at this point, in just a few words, what a 'graffiti wall' is? It doesn't become clear until you get to the images in Figure 1 and lines 55-60.

- Lines 35-29: This is a little confusing and I think a little more care needs to be taken over the logic here. You are making two points, I think, that get conflated: (1) Space is not empty: there is lots of tenuous plasma filling it. (2) There is sound in space: there are pressure waves in plasma. I think it could be worth disentangling the two ideas a little more carefully.

- Line 67: 'The researchers would use what they had written or drawn to prompt a short dialogue about aspects of the space environment [. . .] — a method informed by the 'science capital' research'. The relationship between science capital research and the researchers undertaking a dialogue with attendees is not immediately obvious here. It might be interesting to draw out a couple of details from the research that prompted /

informed this aspect of the activity.

- Line 89-91. The discussion of the power law / Zipf exponent is a little confusing here. In line 89, you say the exponent is -1. However, in lines 90-91, you suggest the exponent can take different values. When is it -1 and when is it something else? Or are these two different things? Some further clarity here would be beneficial.

- Line 124-129. I'm struggling to piece this together a bit. Is the implication that the people who initially said 'empty' then went on to say something else afterwards, but they didn't say 'full'? I think you need to re-examine how you set out your findings here, because it is a bit confusing as it stands.

- Line 201-2. Why do you give 16 responses before and 15 responses after? Overall, the explanation of the contents of Table 2 is hard to follow.

*Technical corrections*

- Line 9: 'the power of data sonification in innately communicating science' – I'm not sure 'innately' is the right word here. I'm not quite sure what you mean.

- Line 35: 'The solar wind is highly dynamic and as it buffets against Earth's magnetic field generates plasma wave analogues to ordinary sound at ultra-low frequencies' – this is difficult to follow as it stands. Consider putting an extra 'it' in: 'as it buffets against Earth's magnetic field, it generates plasma wave analogues'

---

## Author Comment (AC2) · 26 Nov 2020

**\*General comments\* This is a good paper that presents a useful approach to evaluating drop-in public engagement activities. The detailed statistical analysis is particularly interesting, perhaps more for its explication of a rigorous analysis of graffiti walls and word clouds than for its demonstration of the efficacy of this particular activity. The analysis is very impressive and this paper stands to be a constructive best-practice guide for other public engagement practitioners.**

[Figure]

We thank the reviewer for their time in assessing the manuscript and have taken their comments into account with the following responses.

**Nevertheless, I think the novelty of using before-and-after graffiti walls is perhaps overstated. For example, I was part of an interactive drop-in exhibit in March 2018, where we asked attendees to write words / ideas related to the exhibit theme on small cards, both on entry and just prior to leaving, giving us both pre- and post- data, in the form of collections of words and phrases, in much the same way. However, I think the subsequent analysis of data performed here is what makes this work noteworthy, and, as far as I am aware, original.**

We do not doubt that others may have had the idea to put graffiti walls both before and after an activity, however in our literature search we have found no published evidence of this. This is likely because practitioners often do not share their evaluations publicly and thus the learning which develops in science communication and public engagement does not get passed on effectively. This is why journals like Geoscience Communication are important. We rephrase lines 74-75 to "we are unaware of any *published* public engagement activity that has captured *and analysed* data both before and after a drop-in activity using them" to clarify this position. We thank the reviewer for their comments on the novelty of the analysis of the captured before and after data.

**The title adequately reflects the contents of the paper, and the abstract gives a neat summary too. Overall, the paper is well-structured and clear, and of an appropriate length for the material covered. The language is fluent and precise, although there are one or two points (as noted in the specific comments below) where the readability drops off a bit and it becomes confusing. Nevertheless, this paper is largely well-written, useful and enjoyable to read. It makes a worthwhile contribution to the literature of this field.**

We thank the reviewer for these comments.

**\*Specific comments\* - Line 20: Is it worth explaining at this point, in just a few**

**words, what a 'graffiti wall' is? It doesn't become clear until you get to the images in Figure 1 and lines 55-60.**

We will add here that "Graffiti walls are large areas (often a wall, whiteboard, or large piece of paper) where participants are free to write or draw responses in reaction to the engagement activity or some prompt question."

**- Lines 35-29: This is a little confusing and I think a little more care needs to be taken over the logic here. You are making two points, I think, that get conflated: (1) Space is not empty: there is lots of tenuous plasma filling it. (2) There is sound in space: there are pressure waves in plasma. I think it could be worth disentangling the two ideas a little more carefully.**

We will rephrase this paragraph as follows:

> The presence of a medium in space allows for plasma wave analogues to ordinary sound (pressure waves) that occur at ultra-low frequencies — fractions of milliHertz up to 1 Hz — and are routinely measured by space missions. One way in which these waves are generated is through the highly dynamic solar wind buffetting against Earth's magnetic field, a process that plays a key role within space weather and thus how phenomena from space can affect our everyday lives (e.g. Keiling et al., 2016). However, the belief by the public that space is completely empty in turn leads many to incorrectly think that there is absolutely no sound in space, reinforced by school science demonstrations such as the bell-jar experiment (see Caleon et al., 2013, for a nuanced discussion) or even popular culture like in the marketing to the movie 'Alien'. Public engagement with this research area is thus needed to help correct this fallacy.

**- Line 67: 'The researchers would use what they had written or drawn to prompt a short dialogue about aspects of the space environment - a method informed**

**by the 'science capital' research'. The relationship between science capital re-
search and the researchers undertaking a dialogue with attendees is not imme-
diately obvious here. It might be interesting to draw out a couple of details from
the research that prompted / informed this aspect of the activity.**

This brevity was due to the word limit of the GC letter format. We will expand the
discussion of this link between the 'science capital' research and how it informed this
aspect of the activity. These stem from the issue of whether people feel included in
science and that it is for "people like me". The 'science capital' researchers recommend
using and valuing participants' own experiences as part of engagements instead of the
typical transmissive approach which can alienate lower science capital audiences from
the scientists who are trying to engage.

**- Line 89-91. The discussion of the power law / Zipf exponent is a little confusing
here. In line 89, you say the exponent is -1. However, in lines 90-91, you suggest
the exponent can take different values. When is it -1 and when is it something
else? Or are these two different things? Some further clarity here would be
beneficial.**

While the Zipf exponent is typically quoted as -1, it can indeed vary as we later indicate
with further references and the more generalised form is that of a power law. We will
clarify both these points in the text.

**- Line 124-129. I'm struggling to piece this together a bit. Is the implication
that the people who initially said 'empty' then went on to say something else
afterwards, but they didn't say 'full'? I think you need to re-examine how you set
out your findings here, because it is a bit confusing as it stands.**

Again the concise nature of this paragraph was driven by the word limit of the GC letter
format. We will expand the discussion to make this much clearer, but essentially the
reviewer is correct.

**- Line 201-2. Why do you give 16 responses before and 15 responses after? Overall, the explanation of the contents of Table 2 is hard to follow.**

This was because ties in the ranks of words made it impossible to select a suitable subset of equal size in both datasets for the reliability testing. Nonetheless, the log-linear analysis (like a chi-square test) does not require equally sized datasets. We will, however, clarify this as:

> To ensure the reliability of the main qualitative coding of the entire dataset, second coders applied the thematic analysis to a subset of the data. This subset constituted the top 16 words before (58% of total responses) and 15 words after (49%), with the slightly different number of words used in the two datasets being due to ties in the ranking of words making it impossible to have exactly the same number in both. Table B2 shows the totals of how these unique words were grouped across all three coders. These results are used in the log-linear analysis to test reliability, which we note does not require equally sized datasets. The codes' association to the raw data can be found in the supplementary material, both for the main and second coders.

We will also rephrase the caption to the Table.

**\*Technical corrections\* - Line 9: 'the power of data sonification in innately communicating science' – I'm not sure 'innately' is the right word here. I'm not quite sure what you mean.**

Again due to word limits we were not able to further elaborate on this. Given that the words on the graffiti wall afterwards were purely based on participants' reflections to their experience, their innate sense of sound was able to convey key aspects of the science to them before they even spoke to the researchers. We will clarify this in the revised manuscript.
**- Line 35: 'The solar wind is highly dynamic and as it buffets against Earth's magnetic field generates plasma wave analogues to ordinary sound at ultra-low frequencies' – this is difficult to follow as it stands. Consider putting an extra 'it' in: 'as it buffets against Earth's magnetic field, it generates plasma wave analogues'**

We will make this correction.

---

## Referee Comment (RC3) · Victoria Engelschiøn (Referee) · 29 Nov 2020

The authors have written an interesting paper on quantifying visitors' experience of a soundscape exhibit. To evaluate learning outcomes is crucial for designing and developing new and better exhibitions. The authors successfully quantify changes in the visitors' perception before and after the exhibit. I therefore think the purpose of this study is important, and that the study is appropriate for Geoscience Communication. Overall, the writing is short and concise. The structure is logical and flows well. Some sentences are a bit long, and could be shortened for increased readability.

[Figure]

The exhibit is a soundscape and the evaluation is done by using two methods of statistical analyses. Input from the visitors is collected using sticky notes (termed graffiti walls). To me, there are two main points in this study: 1) the use of statistical analyses on engagement reviews and 2) measuring the learning outcomes in a soundscape. I think the statistical analyses have been well-covered by the other reviewers. However, I would have liked to see the authors place their findings more in the context of the exhibit. The intended and the measured learning outcomes seem somewhat detached, while the paper raises several interesting questions regarding the learning outcome. E.g. if the exhibit aimed to teach visitors about plasma waves or space weather, in what way did the authors capture that? Measuring "change" is vague, and I think it should be specified what kind of change they were looking for. This would also be important knowledge for others in the future when deciding on methods to apply. I had some questions whether graffiti walls are accurate enough to adequately capture details in the visitors' perception. In general, space is empty, slow and silent. Is there a risk here that the visitors mixed near-Earth space weather and conditions in outer space? Or that plasma waves are sound waves? Not all change is positive, so would there be any way the authors could measure this in their method? The bell-jar experiment was mentioned as an example that people falsely think space is silent. However, my understanding of the bell-jar issue is that people think only air propagate sound, and that space is silent because there is no air. That waves propagate in plasma, and that these waves can be sonified to be audible for humans, is very complex information. To make sure that visitors did not confuse any of these concepts seems to require targeted questions from the evaluators? The authors' reflections and insights on this would be appreciated. I was not familiar with the term graffiti walls for sticky notes, this should be explained. It would also be interesting with a brief explanation of why this method was chosen. The term young families is not defined, but I assume these are young children and that many of those cannot write? If adults write for them, would this bias the responses to e.g. show higher vocabulary complexity? Line 91 states that Zipf's shows different trends for children and adults. The analysis using Zipf's is

presented for the entire dataset. How would the age distribution affect the result, and could shifts in the age distribution before/after affect these? I was wondering whether the increased diversity in words afterwards, but fewer respondents, could be caused by a larger proportion of adults participating (e.g. because the children were too tired?). Some clarifications or reflections on this would be helpful.

Line 50: Museum is misspelt ("Musueum")

---

## Author Comment (AC3) · 4 Dec 2020

The authors have written an interesting paper on quantifying visitors' experience of a soundscape exhibit. To evaluate learning outcomes is crucial for designing and developing new and better exhibitions. The authors successfully quantify changes in the visitors' perception before and after the exhibit. I therefore think the purpose of this study is important, and that the study is appropriate for Geoscience Communication. Overall, the writing is short and concise. The structure is logical and flows well. Some sentences are a bit long, and could be shortened

[Figure]

**for increased readability.**

We thank the reviewer for their time in constructing comments and have considered them all, with our responses given below.

**The exhibit is a soundscape and the evaluation is done by using two methods of statistical analyses. Input from the visitors is collected using sticky notes (termed graffiti walls). To me, there are two main points in this study: 1) the use of statistical analyses on engagement reviews and 2) measuring the learning outcomes in a soundscape. I think the statistical analyses have been well-covered by the other reviewers. However, I would have liked to see the authors place their findings more in the context of the exhibit. The intended and the measured learning outcomes seem somewhat detached, while the paper raises several interesting questions regarding the learning outcome. E.g. if the exhibit aimed to teach visitors about plasma waves or space weather, in what way did the authors capture that?**

The development of a high quality public engagement activity should ideally be defined by its purpose. These do not necessarily have to be linked to learning specific information, but can encompass many possible intended outcomes (see https://www.artscouncil.org.uk/measuring-outcomes/generic-learning-outcomessection-1 for a helpful framework of describing the myriad of potential outcomes in informal learning and public engagement). The purpose of this activity was to provide young children and their parents (as key influencers) an accessible and immersive experience that would enable participation and spark discussion. Such experiences can contribute to children building an association and identity with science, a key part of a person's 'science capital'. This could not be fully explored due to the word limits of the GC Letters format, but we will clarify our position on the purpose of this exhibit in the revised manuscript as follows:

The purpose of the space soundscape was to provide young children and

their parents/carers (as key influences upon them) an accessible and immersive experience with space research that would enable participation and spark discussion. Such experiences may, when taken in conjunction with all the other formal and informal interactions with science afforded to a young person, contribute towards developing their science identity and hence build their 'science capital' (Archer and DeWitt, 2017). Using a generic learning outcomes framework (Hooper-Green, 2004), the main intentions for the activity fall within the realms of 'Enjoyment, Inspiration, Creativity' and 'Attitudes  Values', with 'Knowledge  Understanding' being only a secondary aim.

**Measuring "change" is vague, and I think it should be specified what kind of change they were looking for. This would also be important knowledge for others in the future when deciding on methods to apply.**

As stated in the introduction, demonstrating impact requires some measure of change. While it is possible to have a very specific change, and thus impact, in mind and thus only evaluate for that, we felt that in this case such an approach was too reductive. Furthermore, given the challenges in evaluating impact at all for drop-in activities generally, as outlined in the introduction, we therefore felt it was better to be open to any sorts of changes that might have resulted from before to after, as we certainly didn't feel we could predict all possible responses in advance. Our approach thus took a more exploratory and data-driven view of the qualitative data capture and analysis. The grounded theory approach of thematic analysis, for example, exemplifies this as it looks for patterns that emerge from the qualitative data itself, as outlined on lines 105-106 and in Appendix B, rather than only looking at the data with a very specific lens.

**I had some questions whether graffiti walls are accurate enough to adequately capture details in the visitors' perception. In general, space is empty, slow and**

**silent. Is there a risk here that the visitors mixed near-Earth space weather and conditions in outer space?**

The graffiti wall provides an open opportunity for participants to reflect upon and respond with their own perceptions and associations with space, a point we will add to the manuscript. The benefits of graffiti walls as evaluative tools are provided in the references contained within the introduction. This method was chosen specifically due to its suitability for evaluating drop-in activities, ability to be integrated within the activity itself, and alignment with our intended overall experience. We will add these points to the manuscript.

We are unsure of exactly where the reviewer is referring to with the term "outer space" since this technically applies to everywhere above 100km altitude. The satellite measurements used here are taken from geostationary orbit, within Earth's magnetosphere. However, similar dynamics and waves are present throughout the entire heliosphere, the Sun's region of influence due to its solar wind (which streams at several hundreds of kilometres a second), which is highlighted on lines 30-34. All stars have their own stellar winds, again leading to similar conditions at other stellar systems. Finally, the interstellar medium is another example of a space plasma, which is in fact denser than the outer regions of the heliosphere as confirmed when Voyager 1 crossed the heliopause in 2012. Therefore, we do not see much risk here as space plasmas are ubiquitous throughout the universe. We will briefly mention these other space plasmas. While it is always possible for participants to draw incorrect conclusions from any activity or form of communication, the activity was carefully designed to avoid this, e.g. with the placement of researchers at the end of the research to enter into dialogues with participants.

**Or that plasma waves are sound waves?**

While not all plasma waves are equivalent to sound (with high frequency plasma waves being driven by the electric fields between ions and electrons kinetically), the ultra-low

frequency waves concerned in this paper are. This is because they, like sound waves in a gas, arise from the fluid (magneto)hydrodynamic equations. The only difference is that in plasmas magnetic effects, such as magnetic pressure, are also included whereas these are not present in a fluid consisting of electrically neutral particles. Nonetheless, a sound wave which propagates from a neutral gas to a magnetised plasma will become a magnetosonic plasma wave. Ultra-low frequency plasma waves are thus even more analogous to sound than even seismic waves, where the medium is not a fluid and the wave propagates due to stresses (via the inter-atomic and inter-molecular bonds present) rather than simply pressure, despite many members of the public being comfortable equating seismic waves to sound. Therefore, we again do not see major issues here. We will add a reference to an article (Archer, M. O.: In space no-one can hear you scream. . .or can they?, ENT Audiology News, Volume 28, Issue 6, 2020) which discusses these aspects about the nature of the plasma waves in the context of sound and other waves, as we feel such a discussion detracts from the point of this paper.

**Not all change is positive, so would there be any way the authors could measure this in their method?**

The analysis could indeed have captured negative impacts. The quantitative linguistics could have revealed a decreased diversity of words following the soundscape. Additionally, the changes in the qualitative codes might have shown an increase in codes related to misconceptions about space rather than a decrease. Finally, the generation of the qualitative codes drawn from the data itself, rather than using preconceived themes/codes, could have highlighted negative themes. None of these were found, however. We can point this out in the paper.

**The bell-jar experiment was mentioned as an example that people falsely think space is silent. However, my understanding of the bell-jar issue is that people think only air propagate sound, and that space is silent because there is no air.**

The misconception with the bell-jar experiment is particularly related to the "vacuum", as a bell-jar never becomes completely devoid of air. This is discussed in the referenced paper of Caleon et al. (2013), presenting a more nuanced description of the experiment in near-vacuum conditions and how it should ideally be presented. We will highlight this slightly more in the manuscript. As to sound requiring air, many school curricula discuss the propagation of sound through other mediums, such as water. Indeed, most people will be aware that you can hear sound underwater from swimming. The misconception the reviewer describes is something we have never encountered.

**That waves propagate in plasma, and that these waves can be sonified to be audible for humans, is very complex information. To make sure that visitors did not confuse any of these concepts seems to require targeted questions from the evaluators? The authors' reflections and insights on this would be appreciated.**

The reviewer seems to have assumed learning objectives surrounding the concept of the exhibit itself. However, this was not the case as highlighted in our previous response about its purpose. The complex information/discussions that the reviewer describes were generally not warranted. It is clear from the changes in the qualitative codes before and directly after the soundscape that simple concepts of space not being empty, sound being present, dynamics occurring, and electricity being present were innately communicated to the participants simply through listening to the data. It was only these sorts of simple messages that would have been reinforced by the researchers in their dialogues afterwards. We will note that the reviewer interactions were specifically designed to cement or clarify conceptions in a tailored and audience-focused way, e.g. only going into an appropriate level of detail depending on the individual or group.

The series of targeted questions that the reviewer suggests would have run contrary to best practice in the evaluation of drop-in activities, as outlined in the introduction, since they would not have been commensurate with the activity and would risk interfering with participants' experience. As highlighted in our previous responses, we will clarify why we chose this method of evaluation and its benefits in this context.
**I was not familiar with the term graffiti walls for sticky notes, this should be explained. It would also be interesting with a brief explanation of why this method was chosen.**

We will add a description of a graffiti wall in the introduction.

**The term young families is not defined, but I assume these are young children and that many of those cannot write? If adults write for them, would this bias the responses to e.g. show higher vocabulary complexity? Line 91 states that Zipf's shows different trends for children and adults. The analysis using Zipf's is presented for the entire dataset. How would the age distribution affect the result, and could shifts in the age distribution before/after affect these? I was wondering whether the increased diversity in words afterwards, but fewer respondents, could be caused by a larger proportion of adults participating (e.g. because the children were too tired?). Some clarifications or reflections on this would be helpful.**

Young families is a common term for families with young children. For ethical reasons we did not collect personal characteristics from participants, as stated on lines 76-77, therefore we purposefully do not try to give specific age ranges for those that might have attended. Observations did not highlight that adults were largely writing on behalf of their children, as younger children had to the option to draw as well as write. In fact, as noted on lines 78-79, it was observed that in families typically only the children contributed to the graffiti walls rather than the adults. While one might expect different absolute values of the Zipf exponents if the data could be subdivided by age, here we are interested only in changes to the Zipf exponent from before to after rather than the exponent's specific value. The changes presented, however, are robust since we observed no substantive difference in those filling in the graffiti walls before or after the activity. Furthermore, the number of responses show that the vast majority of respondents (83%) participated in both graffiti walls. The exhibit, as a drop-in activity, lasted mere minutes and we saw no evidence of children becoming tired due to it. We

will clarify all of these points in the paper.

**Line 50: Museum is misspelt ("Musueum")**

We will correct this.

---

## Author Comment (AC4) · 9 Dec 2020

In addition to our previous response, which all co-authors agree with, we have subsequently also performed a Kolmogorov-Smirnov test on the two datasets used in the Zipf exponent analysis. This shows that the two entire distributions are significantly different ($p = 8 \times 10^{-11}$), further backing up our result of an increased diversity of words afterwards compared to before.
* * *

---

## Author Response (AR1)

**Response to reviewers**

'Demonstrating change from a drop-in engagement activity through pre- and post-graffiti walls: Quantitative linguistics and thematic analysis applied to a space soundscape exhibit'

Archer et al.

We thank the editor and the reviewers for their comments. We have revised the manuscript in response to these, which we detail here. Line numbers refer to the tracked changes version of the manuscript.

**Editor comments**

**Firstly, thank you to the reviewers who have provided some very constructive and useful feedback. And thank you to the authors for responding in a respectful way. Your responses reassure me that the paper will be updated in a way that responds to the reviewers questions and suggestions.**

We thank the editor for their comments.

**I have a few points which I would also like to add. Some of the following are centred around the reviewers points. Other things are based on my own reading of the article. Please consider the following:**

**It would be great if you could add a short description of the "science capital" approach that you mention as the framework behind the exhibit.**

We have added on line 74 "`Science capital' is defined as the total science-related knowledge, attitudes, experiences, and resources that a person has built up over their life (Archer and DeWitt, 2017). This includes what science they know about, what they think and feel about science, the people they know and their relation to science, and the day-to-day engagement they have with science."

**I would appreciate a brief explanation of the ranking of the words in section 4.1. What does this ranking mean? Does it mean that the more a word is used the higher it's rank (i.e closer to 1)? If so please, put this in.**

The editor is correct in their understanding of the ranks and we have added this to the paper on line 138.

**At a more general level, I would like you to describe what you mean by "noisy". As I am sure you agree, "noisy" can mean different things. If I was standing on top of a mountain in the middle of nowhere, I would likely describe it as "silent". However, I am sure there are sound waves all around me that can be sonified to make that mountain top much more "noisy" than I first imagined. Is this the same with space? Is it only "noisy" if we sonifiy the waves? If so, I think it could be useful to explain this. Space isn't "noisy" in relation to what we think here on Earth, but it's "noisy" in so far as there are sound waves that we can sonify and then hear. I hope you see my point here.**

The editor raises an interesting point. We have added a brief discussion of two ways in which space can be considered "noisy" – relative to prior expectations thanks to the sonification (as the editor intimated), but also relative to the background pressure. This can be found on lines 183-191.

**In your response to reviewer 2, you add a description of what a "graffiti wall" is in relation to your project. This is very useful. However, it brings up a limitation with your project which I would like you to comment on in the paper. You say that graffiti walls are large areas where "participants are free to write and draw responses…". In your paper you only evaluate the words people use and not their drawings. I think it is important that you note whether there were drawings on the post-it notes and briefly say that these are not evaluated here (and maybe that this sort of analysis needs to use different methods for evaluation).**

The editor is correct in that we did not evaluate the drawings in the quantitative linguistics part of the analysis, for obvious reasons, and we have now highlighted this in section 4.1 explicitly on line 145-146. However, we did include the drawings within the thematic analysis, as was noted on line 162.

**With regard to Reviewer Engelschiøn's comment on "Measuring "change" is vague". Your response is good, but I believe there is room in the discussion part of this paper for you to expand a little on whether you see the change as positive or negative. Indeed you say that the approach took "a more exploratory and data-driven view". However, you are experts in space science. You can surely say something about whether you feel the change you noted was good or bad. In my mind, the change seems very positive, but it would be nice to hear what you think. It is great that you follow up on this issue up with the same reviewer on page C5. I look forward to reading your additions on this.**

We agree with the editor that the changes from before to after the soundscape do appear to indicate positive impacts on participants. We have added comments to this effect when discussing the results (lines 157, 222, 237-240, and 248-249). We also note that the co-authors are engagement specialists with no background in space science, which lends further confidence to the impacts reported.

**Finally, and importantly. It would be nice to see a comment on any ethical considerations. You evaluated input from human subjects, where you should consider ethics. It is quite possible that the ethical guidelines at your university do not apply to your specific methodology. However, it would be very useful to mention that you considered this and why. You have already done a fantastic job implementing and describing the evaluation of your communication activity. To include the ethical guidelines will make your paper a very nice example of what geoscientists should be aiming for when planning, conducting and writing up their communication activities.**

We thank the editor for this suggested addition. We note on lines 117-122 that "Ethical considerations in the design of the exhibit and its evaluation followed the BERA (2018) guidelines and were discussed with institutional funders and the Science Museum before the activity occurred. All respondents consented to providing graffiti wall responses as these were not mandatory for participation in the soundscape exhibit. Children only participated in any of the activities when consented and accompanied by their appropriate adult. All data collected was anonymous and no characteristics about participants were solicited. Due to the nature of the exhibit, its design, and the responses being collected, it was deemed that there was very little risk of harm arising from participation."

**RC1**
**In general I enjoyed this manuscript and feel it would be suitable for publication in Geoscience Communication with some changes which I outline below.**

We thank Dr Coxon for their time in promptly assessing the manuscript and have considered their comments carefully.

**95–99: I was unfamiliar with Zipf's law, and I think applying this to evaluate public engagement in this manner is highly novel and very impressive. The less negative the exponent, the wider the vocabulary. In Figure 3, this clearly demonstrates that for the ~10 highest ranks, there was a significantly wider range of vocabulary. However, I do have some concerns about things that are unclear. Firstly, I think it is necessary to express and discuss the percentage of the words captured in the ~10 highest ranks; from an inspection of the graph it looks to me like ~90% for the "before" set and ~80% for the "after" set, so the vast majority of the language used on the walls is presumably contained in those 10 upper ranks (if I'm misinterpreting this graph, quoting these numbers explicitly is even more important).**

The proportion of words captured within a segment is the sum of the relative frequencies within that segment, which is similar to the area under the graph. We thank the reviewer for their suggestion to include the percentages in each segment, which can be found on line 154.

**Secondly, the authors say "While the exponents in the lowest ranked segments are consistent with one another," but the two exponents are not consistent with one another; -1.00\pm0.04\neq-0.85\pm0.05. This part needs to be rewritten to acknowledge that a) although the two are similar, in fact the variety of language at rank > 10 decreased after the study compared to before; and b) what implications this has.**

This statement was based on the fact that the two segments' error bars overlapped each other's regression lines, however, the reviewer's point has raised the fact that we did not propagate through these errors in the proportions into our standard error in the slope. Doing this we arrive at the exponents in the revised Figure 3. This now better illustrates that the exponents in the lower ranked segments are indeed consistent with one another as well as the standard Zipf exponent of -1. Thus the implication is that there was significantly increased diversity of language in approximately half the dataset, whereas the other half were consistent with one another, thereby making the overall effect positive. We have also performed a Kolmogorov-Smirnov test on the two datasets. This shows that the two entire distributions are significantly different ($p = 8 \times 10^{-11}$), further backing up our result of an increased diversity of words afterwards compared to before, as now noted on line 155.

**105: The authors say that "instead of using pre-determined qualitative codes, the analysis drew on grounded theory". I am unfamiliar with the word "code" used in this context. Appendix B particularly, but also the citations, helped me grasp what the meaning is (i.e., for each word, identifying its theme and then within that theme whether it means one of two extremes, i.e. "quiet" or "loud", "empty" or "full", and "slow or busy"). However, I am concerned that readers might not be as tenacious in reading around the terminology as I was as a reviewer, and so I think the paper would be much improved with a fuller explanation here.**

We thank the reviewer for this suggestion. We now include the "This finds patterns, known as qualitative codes, in the data which are then grouped into broader related themes" after the introduction of thematic analysis on line 163. We also highlight that in our scheme the codes within a theme are typically antithetical.

**108–112: Tying into my criticisms above, a better way to express this would be a table which explicitly outlines which of these are themes and which are codes, I think.**

We agree with the reviewer and have added Table 1 in response.

**113: I had not heard of Cronbach's alpha prior to this manuscript. Most of the examples I have found of its use during my reading for this review refer to applying it to Likert scaled quantities to compare the extent to which, say, a questionnaire of questions about a given theme are telling the author about that theme. This indicates two things; firstly, to apply it the themes must have been assigned numerical values, but it is not clear to me what those values were from the explanation herein. If the authors use Cronbach's alpha, they need to explain in more detail how the themes were quantised in order to apply the measure (are these simply the 1/2 numbers expressed in Figure 4? If so, say that, and if not, then what are they?). Secondly, as far as I can tell from my exploration of the literature, Cronbach's alpha is applied to measure reliability, not dimensionality (as in this study); in fact, some literature I read specifically cautioned against the latter and the Cho paper cited does not seem to provide a rationale for using the measure in this way. I have no real issue with the central analysis tenet in Section 4.2, namely that the coder can differentiate to which theme a word belongs. However, if I did have an issue with it, the quoted statistic would not convince me, and should either be removed or the reasons for its validity should be expanded on significantly.**

The reviewer raises valid concerns on the regular usage of Cronbach's alpha within qualitative research in general and we appreciate that due to the word limit of the GC letters format its usage was not clearly discussed. Given that Cronbach's alpha is not critical to the methods or results in the manuscript, we feel it best to simply remove this sentence.

**RC2**

**\*General comments\***

**This is a good paper that presents a useful approach to evaluating drop-in public engagement activities. The detailed statistical analysis is particularly interesting, perhaps more for its explication of a rigorous analysis of graffiti walls and word clouds than for its demonstration of the efficacy of this particular activity. The analysis is very impressive and this paper stands to be a constructive best-practice guide for other public engagement practitioners.**

We thank the reviewer for their time in assessing the manuscript and have taken their comments into account with the following responses.

**Nevertheless, I think the novelty of using before-and-after graffiti walls is perhaps overstated. For example, I was part of an interactive drop-in exhibit in March 2018, where we asked attendees to write words / ideas related to the exhibit theme on small cards, both on entry and just prior to leaving, giving us both pre- and post- data, in the form of collections of words and phrases, in much the same way. However, I think the subsequent analysis of data performed here is what makes this work noteworthy, and, as far as I am aware, original.**

We do not doubt that others may have had the idea to put graffiti walls both before and after an activity, however in our literature search we have found no published evidence of this. This is likely because practitioners often do not share their evaluations publicly and thus the learning which develops in science communication and public engagement does not get passed on effectively. This is why journals like Geoscience Communication are important. We rephrase lines 74-75 to "we are unaware of any *published* public engagement activity that has captured *and analysed* data both before and after a drop-in activity using them" to clarify this position (see revised manuscript lines 114). We thank the reviewer for their comments on the novelty of the analysis of the captured before and after data.

**The title adequately reflects the contents of the paper, and the abstract gives a neat summary too. Overall, the paper is well-structured and clear, and of an appropriate length for the material covered. The language is fluent and precise, although there are one or two points (as noted in the specific comments below) where the readability drops off a bit and it becomes confusing. Nevertheless, this paper is largely well-written, useful and enjoyable to read. It makes a worthwhile contribution to the literature of this field.**

We thank the reviewer for these comments.

**\*Specific comments\***

**- Line 20: Is it worth explaining at this point, in just a few words, what a 'graffiti wall' is? It doesn't become clear until you get to the images in Figure 1 and lines 55-60.**

We have added on lines 30-32 that "Graffiti walls are large areas (often a wall, whiteboard, or large piece of paper) where participants are free to write or draw responses in reaction to the engagement activity or some prompt question, either directly on the area itself or by sticking responses to it."

**- Lines 35-29: This is a little confusing and I think a little more care needs to be taken over the logic here. You are making two points, I think, that get conflated: (1) Space is not empty: there is lots of tenuous plasma filling it. (2) There is sound in space: there are pressure waves in plasma. I think it could be worth disentangling the two ideas a little more carefully.**

We have significantly rephrased this paragraph, see lines 45-61.

**- Line 67: 'The researchers would use what they had written or drawn to prompt a short dialogue about aspects of the space environment - a method informed by the 'science capital' research'. The relationship between science capital research and the researchers undertaking a dialogue with attendees is not immediately obvious here. It might be interesting to draw out a couple of details from the research that prompted / informed this aspect of the activity.**

This brevity was due to the word limit of the GC letter format. We expand the discussion of this link between the 'science capital' research and how it informed this aspect of the activity on lines 100-105. These stem from the issue of whether people feel included in science and that it is for "people like me". The 'science capital' researchers recommend using and valuing participants' own experiences as part of engagements instead of the typical transmissive approach which can alienate lower science capital audiences from the scientists who are trying to engage.

**- Line 89-91. The discussion of the power law / Zipf exponent is a little confusing here. In line 89, you say the exponent is -1. However, in lines 90-91, you suggest the exponent can take different values. When is it -1 and when is it something else? Or are these two different things? Some further clarity here would be beneficial.**

While the Zipf exponent is typically quoted as -1, it can indeed vary as we later indicate with further references and the more generalised form is that of a power law. We clarify both these points in the text on lines 137-143.

**- Line 124-129. I'm struggling to piece this together a bit. Is the implication that the people who initially said 'empty' then went on to say something else afterwards, but they didn't say 'full'? I think you need to re-examine how you set out your findings here, because it is a bit confusing as it stands.**

Again the concise nature of this paragraph was driven by the word limit of the GC letter format. We expand the discussion to make this much clearer on lines 203-206, but essentially the reviewer is correct.

**- Line 201-2. Why do you give 16 responses before and 15 responses after? Overall, the explanation of the contents of Table 2 is hard to follow.**

This was because ties in the ranks of words made it impossible to select a suitable subset of equal size in both datasets for the reliability testing. Nonetheless, the log-linear analysis (like a chi-square test) does not require equally sized datasets. We clarify this on lines 298-305 and have also rephrased the caption to the Table.

**\*Technical corrections\***

**- Line 9: 'the power of data sonification in innately communicating science' – I'm not sure 'innately' is the right word here. I'm not quite sure what you mean.**

Again due to word limits we were not able to further elaborate on this. Given that the words on the graffiti wall afterwards were purely based on participants' reflections to their experience, their innate sense of sound was able to convey key aspects of the science to them before they even spoke to the researchers. We clarify this in the revised manuscript on lines 11-12.

**- Line 35: 'The solar wind is highly dynamic and as it buffets against Earth's magnetic field generates plasma wave analogues to ordinary sound at ultra-low frequencies' – this is difficult to follow as it stands. Consider putting an extra 'it' in: 'as it buffets against Earth's magnetic field, it generates plasma wave analogues'**

We have made this correction.

**RC3**

**The authors have written an interesting paper on quantifying visitors' experience of a soundscape exhibit. To evaluate learning outcomes is crucial for designing and developing new and better exhibitions. The authors successfully quantify changes in the visitors' perception before and after the exhibit. I therefore think the purpose of this study is important, and that the study is appropriate for Geoscience Communication. Overall, the writing is short and concise. The structure is logical and flows well. Some sentences are a bit long, and could be shortened for increased readability**

We thank the reviewer for their time in constructing comments and have considered them all, with our responses given below.

**The exhibit is a soundscape and the evaluation is done by using two methods of statistical analyses. Input from the visitors is collected using sticky notes (termed graffiti walls). To me, there are two main points in this study: 1) the use of statistical analyses on engagement reviews and 2) measuring the learning outcomes in a soundscape. I think the statistical analyses have been well-covered by the other reviewers. However, I would have liked to see the authors place their findings more in the context of the exhibit. The intended and the measured learning outcomes seem somewhat detached, while the paper raises several interesting questions regarding the learning outcome. E.g. if the exhibit aimed to teach visitors about plasma waves or space weather, in what way did the authors capture that?**

The development of a high quality public engagement activity should ideally be defined by its purpose. These do not necessarily have to be linked to learning specific information, but can encompass many possible intended outcomes (see https://www.artscouncil.org.uk/measuring-outcomes/generic-learning-outcomes#section-1 for a helpful framework of describing the myriad of potential outcomes in informal learning and public engagement). The purpose of this activity was to provide young children and their parents (as key influencers) an accessible and immersive experience that would enable participation and spark discussion. Such experiences can contribute to children building an association and identity with science, a key part of a person's 'science capital'. This could not be fully explored due to the word limits of the GC Letters format, but we now clarify our position on the purpose of this exhibit in the revised manuscript on lines 80-86.

**Measuring "change" is vague, and I think it should be specified what kind of change they were looking for. This would also be important knowledge for others in the future when deciding on methods to apply.**

As stated in the introduction, demonstrating impact requires some measure of change. While it is possible to have a very specific change, and thus impact, in mind and thus only evaluate for that, we felt that in this case such an approach was too reductive. Furthermore, given the challenges in evaluating impact at all for drop-in activities generally, as outlined in the introduction, we therefore felt it was better to be open to any sorts of changes that might have resulted from before to after, as we certainly didn't feel we could predict all possible responses in advance. Our approach thus took a more exploratory and data-driven view of the qualitative data capture and analysis. The grounded theory approach of thematic analysis, for example, exemplifies this as it looks for patterns that emerge from the qualitative data itself, as outlined on lines 105-106 and in Appendix B, rather than only looking at the data with a very specific lens. These points have now been noted on lines 165-167.

**I had some questions whether graffiti walls are accurate enough to adequately capture details in the visitors' perception. In general, space is empty, slow and silent. Is there a risk here that the visitors mixed near-Earth space weather and conditions in outer space?**

The graffiti wall provides an open opportunity for participants to reflect upon and respond with their own perceptions and associations with space, a point we have added to the manuscript (line 30-33). The benefits of graffiti walls as evaluative tools are provided in the references contained within the introduction. This method was chosen specifically due to its suitability for evaluating drop-in activities, ability to be integrated within the activity itself, and alignment with our intended overall experience (see lines 110-113)

We are unsure of exactly where the reviewer is referring to with the term "outer space" since this technically applies to everywhere above 100km altitude. The satellite measurements used here are taken from geostationary orbit, within Earth's magnetosphere. However, similar dynamics and waves are present throughout the entire heliosphere, the Sun's region of influence due to its solar wind (which streams at several hundreds of kilometres a second). All stars have their own stellar winds, again leading to similar conditions at other stellar systems. Finally, the interstellar medium is another example of a space plasma, which is in fact denser than the outer regions of the heliosphere as confirmed when Voyager 1 crossed the heliopause in 2012. Therefore, we do not see much risk here as space plasmas are ubiquitous throughout the universe. We now briefly mention these other space plasmas on lines 47-49. While it is always possible for participants to draw incorrect conclusions from any activity or form of communication, the activity was carefully designed to avoid

this, e.g. with the placement of researchers at the end of the soundscape to enter into dialogues with participants.

**Or that plasma waves are sound waves?**

While not all plasma waves are equivalent to sound (with high frequency plasma waves being driven by the electric fields between ions and electrons kinetically), the ultra-low frequency waves concerned in this paper are. This is because they, like sound waves in a gas, arise from the fluid (magneto)hydrodynamic equations. The only difference is that in plasmas magnetic effects, such as magnetic pressure, are also included whereas these are not present in a fluid consisting of electrically neutral particles. Nonetheless, a sound wave which propagates from a neutral gas to a magnetised plasma will become a magnetosonic plasma wave. Ultra-low frequency plasma waves are thus even more analogous to sound than even seismic waves, where the medium is not a fluid and the wave propagates due to stresses (via the inter-atomic and inter-molecular bonds present) rather than simply pressure, despite many members of the public being comfortable equating seismic waves to sound. Therefore, we again do not see major issues here. We have added a reference to an article (Archer, M. O.: In space no-one can hear you scream…or can they?, ENT & Audiology News, Volume 28, Issue 6, 2020) which discusses these aspects about the nature of the plasma waves in the context of sound and other waves on lines 52-53, as we feel such a discussion detracts from the point of this paper.

**Not all change is positive, so would there be any way the authors could measure this in their method?**

The analysis could indeed have captured negative impacts. The quantitative linguistics could have revealed a decreased diversity of words following the soundscape. Additionally, the changes in the qualitative codes might have shown an increase in codes related to misconceptions about space rather than a decrease. Finally, the generation of the qualitative codes drawn from the data itself, rather than using preconceived themes/codes, could have highlighted negative themes. None of these were found, however. We have mentioned these now on lines 150-152 and 165-167.

**The bell-jar experiment was mentioned as an example that people falsely think space is silent. However, my understanding of the bell-jar issue is that people think only air propagate sound, and that space is silent because there is no air.**

The misconception with the bell-jar experiment is particularly related to the "vacuum", as a bell-jar never becomes completely devoid of air. This is discussed in the referenced paper of Caleon et al. (2013), presenting a more nuanced description of the experiment in near-vacuum conditions and how it should ideally be presented. We highlight this slightly more in the manuscript on lines 58-59. As to sound requiring air, many school curricula discuss the propagation of sound through other mediums, such as water. Indeed, most people will be aware that you can hear sound underwater from swimming. The misconception the reviewer describes is something we have never encountered.

**That waves propagate in plasma, and that these waves can be sonified to be audible for humans, is very complex information. To make sure that visitors did not confuse any of these concepts seems to require targeted questions from the evaluators? The authors' reflections and insights on this would be appreciated.**

The reviewer seems to have assumed learning objectives surrounding the concept of the exhibit itself. However, this was not the case as highlighted in our previous response about its purpose. The

complex information/discussions that the reviewer describes were generally not warranted. It is clear from the changes in the qualitative codes before and directly after the soundscape that simple concepts of space not being empty, sound being present, dynamics occurring, and electricity being present were innately communicated to the participants simply through listening to the data. It was only these sorts of simple messages that would have been reinforced by the researchers in their dialogues afterwards. We have added on lines 103-105 that the researcher interactions were specifically designed to cement or clarify conceptions in a tailored and audience-focused way, e.g. only going into an appropriate level of detail depending on the individual or group.

The series of targeted questions that the reviewer suggests would have run contrary to best practice in the evaluation of drop-in activities, as outlined in the introduction, since they would not have been commensurate with the activity and would risk interfering with participants' experience. As highlighted in our previous responses, we have clarified why we chose this method of evaluation and its benefits in this context.

**I was not familiar with the term graffiti walls for sticky notes, this should be explained. It would also be interesting with a brief explanation of why this method was chosen.**

We have added a description of a graffiti wall in the introduction on lines 30-33.

**The term young families is not defined, but I assume these are young children and that many of those cannot write? If adults write for them, would this bias the responses to e.g. show higher vocabulary complexity? Line 91 states that Zipf's shows different trends for children and adults. The analysis using Zipf's is presented for the entire dataset. How would the age distribution affect the result, and could shifts in the age distribution before/after affect these? I was wondering whether the increased diversity in words afterwards, but fewer respondents, could be caused by a larger proportion of adults participating (e.g. because the children were too tired?). Some clarifications or reflections on this would be helpful.**

Young families is a common term for families with young children. For ethical reasons we did not collect personal characteristics from participants (lines 120-121), therefore we purposefully do not try to give specific age ranges for those that might have attended. Observations did not highlight that adults were largely writing on behalf of their children, as younger children had to the option to draw as well as write. In fact, it was observed that in families typically only the children contributed to the graffiti walls rather than the adults (lines 125-126). While one might expect different absolute values of the Zipf exponents if the data could be subdivided by age, here we are interested only in changes to the Zipf exponent from before to after rather than the exponent's specific value (lines 150-152). The changes presented, however, are robust since we observed no substantive difference in those filling in the graffiti walls before or after the activity. Furthermore, the number of responses show that the vast majority of respondents (83%) participated in both graffiti walls. The exhibit, as a drop-in activity, lasted mere minutes and we saw no evidence of children becoming tired due to it.

**Line 50: Museum is misspelt ("Museuem")**

We have corrected this.

**EC1**

**During a recent virtual writing retreat, we used a peer-review framework to review your abstract. We then had an open discussion and noted down all the feedback. We also reviewed your abstract with the following advice in mind: "The abstract is a condensed and concentrated version of the full text of the research manuscript. It should be sufficiently representative of the paper if read as**

a stand-alone document". We looked for the important elements that we believe should be in a research abstract and we comment on them below. We hope the following is helpful for your revisions.

Overall: We were really interested in your project, this "new approach" of graffiti walls and how you have evaluated them. The graffiti walls are a fun method, and we also really liked how you link methods for analysing vocabulary and illustrations. The word "cool" came up a lot when we discussed your project!

The Abstract contained all the necessary parts, which was very positive. You might want to consider tightening up or re-focussing some of these parts to make the Abstract clearer and more concise.

Title: The title contains a lot of information which is great. However, we hope you can make it shorter and more concise. It seems to put more emphasis on the evaluation analyses you used rather than the innovation of the graffiti walls themselves. Several in the group had to read the title several times to pick up on the message. A couple of people misunderstood and thought that the graffiti wall was within the soundscape itself, and that you tested the soundscape before and after the wall was graffitied on.

Need and relevance: The first sentence conveys the need and relevance of the research. However, please consider editing so that it is clearer. Maybe consider two sentences.

Question/hypothesis: Here we misunderstood whether the focus was the graffiti walls being the evaluation itself or the evaluation of the graffiti walls. Right now, it seems that the evaluation methods of quantitative linguistics and thematic analysis could be the main focus of the research question. However, we feel that the graffiti walls themselves are meant to be the main issue. This confusion probably comes from the use of "analysis", "evaluation", "techniques" and "method" in rather quick succession. You might want to look into this to ensure the focus of your main research question/objective is clearly conveyed.

Methods: The methods used for evaluating the graffiti walls are interesting and novel. It's good that you state both methods in the Abstract, but it's not clear what the methods actually do. This part gets quite confusing since it is technically contained in a 5-line sentence, with several clauses. You might consider splitting the methods and results more clearly, and not containing so much detail.

Results: Your main results should refer back to whether the graffiti walls themselves functioned as an evaluation tool. You may also want to refer back to the aim of the whole exhibit. Yes, the graffiti walls may show change, but do they show change relevant to the aim of the actual exhibit itself? Again, the text gets a little complex at this point especially where you write "dynamism, emptiness and electricity, areas highly relevant to the underlying space plasma physics of the sonified data". Our group got a little lost here and asked whether this level of technicality was necessary in the Abstract. You might consider referring to these terms in the main text where you have more room to explain.

Take-home message: You have some really nice results here, that we think you can convey stronger in the final two sentences. Refer to the graffiti walls specifically again and what potential they have. By writing "more broadly" makes it sound like they are already being used in certain circles. Do you mean "more broadly" within soundscapes exhibits or "more broadly" for all drop-in activities in science communication?

**Clarity: When it comes to clarity and conciseness, we would like to ask you to consider whether there are redundant words in the Abstract that you could delete. We would also like you to try and split some of the complex sentences to help with the flow of the story.**

**Spelling/grammar: We all had an issue with the use of "pre- and post-X" with no noun after, in both the title and the text. We're pretty sure you did not mean pre- and postgraffiti walls. If you meant pre- and post-activity graffiti walls, that makes more sense. But you may want to simply use "before and after" as you do later in the text. That's much easier for the reader (us) to relate to.**

**Again, this seems like a really innovative and exciting project. We hope our comments and suggestions help to make the Abstract even better.**

We thank Dr Stiller-Reeve and the members of The Norwegian Research School for Dynamics and Evolution of Earth and Planets (DEEP) for their review of our manuscript's title and abstract. It raised several helpful points which we have taken into account to improve not only these aspects, but the entire manuscript. We have simplified the title to "Demonstrating change from a drop-in space soundscape exhibit by using graffiti walls both before and after". We have also revised the abstract in response.

---

## Author Response (AR2)

**Response to Editor**

**Thank you for taking the time to respond politely and constructively to the reviewers and to update you paper. I am very happy with the overall result. I will submit the paper to be published after a couple of small revisions.**

We thank the editor for their comments on the revisions.

**1. Please be consistent with the tense you use when describing the activity at the science exhibit on page 4. At present there is a mix of present and past tense. I would recommend that all the 4 points are placed in the past/past perfect tense, since everything you describe happened in the past.**

We have changed the tense to be consistent using the editor's suggestion.

**2. On line 108, please provide the full name for the BERA acronym.**

We have expanded the acronym.

**Once these changes are completed, the paper will proceed to the publishing phase. Thanks again for constructive review process.**